# Neural Networks for Efficient Bayesian Decoding of Natural Images from Retinal Neurons

**Nikhil Parthasarathy**[*]
Stanford University
nikparth@gmail.com

**Eleanor Batty**[*]
Columbia University
erb2180@columbia.edu

**William Falcon**
Columbia University
waf2107@columbia.edu

**Thomas Rutten**
Columbia University
tkr2112@columbia.edu

**Mohit Rajpal**
Columbia University
mr3522@columbia.edu

**E.J. Chichilnisky**[†]
Stanford University
ej@stanford.edu

**Liam Paninski**[†]
Columbia University
liam@stat.columbia.edu

## Abstract

Decoding sensory stimuli from neural signals can be used to reveal how we sense our physical environment, and is valuable for the design of brain-machine interfaces. However, existing linear techniques for neural decoding may not fully reveal or exploit the fidelity of the neural signal. Here we develop a new approximate Bayesian method for decoding natural images from the spiking activity of populations of retinal ganglion cells (RGCs). We sidestep known computational challenges with Bayesian inference by exploiting artificial neural networks developed for computer vision, enabling fast nonlinear decoding that incorporates natural scene statistics implicitly. We use a decoder architecture that first linearly reconstructs an image from RGC spikes, then applies a convolutional autoencoder to enhance the image. The resulting decoder, trained on natural images and simulated neural responses, significantly outperforms linear decoding, as well as simple point-wise nonlinear decoding. These results provide a tool for the assessment and optimization of retinal prosthesis technologies, and reveal that the retina may provide a more accurate representation of the visual scene than previously appreciated.

## 1 Introduction

Neural coding in sensory systems is often studied by developing and testing encoding models that capture how sensory inputs are represented in neural signals. For example, models of retinal function are designed to capture how retinal ganglion cells (RGCs) respond to diverse patterns of visual stimulation. An alternative approach – decoding visual stimuli from RGC responses – provides a complementary method to assess the information contained in RGC spikes about the visual world [31, 37]. Understanding decoding can also be useful for the design of retinal prostheses, by providing a measure of the visual restoration that is possible with a prosthesis [26].

The most common and well-understood decoding approach, linear regression, has been used in various sensory systems [29, 40]. This method was shown to be successful at reconstructing white noise temporal signals from RGC activity [37] and revealed that coarse structure of natural image patches could be recovered from ensemble responses in the early visual system [33]. Other linear methods

---

[*],[†]Equal contributions

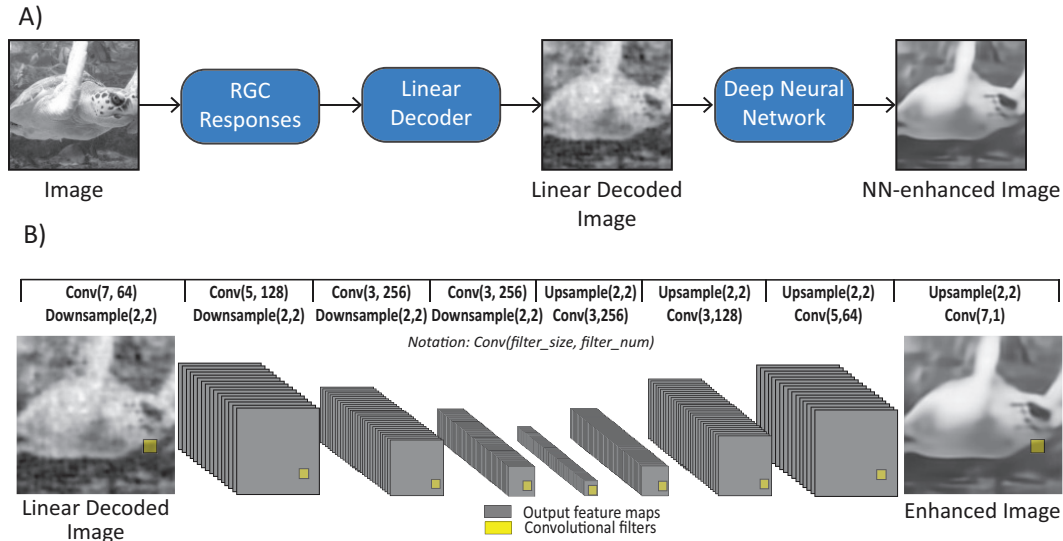

Figure 1: Outline of approach. A) The original image is fed through the simulated neural encoding models to produce RGC responses on which we fit a linear decoder. A deep neural network is then used to further enhance the image. B) We use a convolutional autoencoder with a 4 layer encoder and a 4 layer decoder to enhance the linear decoded image.

such as PCA and linear perceptrons have been used to decode low-level features such as color and edge orientation from cortical visual areas [14, 4]. For more complex natural stimuli, computationally expensive approximations to Bayesian inference have been used to construct decoders that incorporate important prior information about signal structure [25, 27, 30]. However, despite decades of effort, deriving an accurate prior on natural images poses both computational and theoretical challenges, as does computing the posterior distribution on images given an observed neural response, limiting the applicability of traditional Bayesian inference.

Here we develop and assess a new method for decoding natural images from the spiking activity of large populations of RGCs, to sidestep some of these difficulties[1] . Our approach exploits inference tools that approximate optimal Bayesian inference, and emerge from the recent literature on deep neural network (DNN) architectures for computer vision tasks such as super-resolution, denoising, and inpainting [17, 39]. We propose a novel staged decoding methodology – linear decoding followed by a (nonlinear) DNN trained specifically to enhance the images output by the linear decoder – and use it to reconstruct natural images from realistic simulated retinal ganglion cell responses. This approach leverages recent progress in deep learning to more fully incorporate natural image priors in the decoder. We show that the approach substantially outperforms linear decoding. These findings provide a potential tool to assess the fidelity of retinal prostheses for treating blindness, and provide a substantially higher bound on how accurately real visual signals may be represented in the brain.

## 2   Approach

To decode images from spikes, we use a linear decoder to produce a baseline reconstructed image, then enhance this image using a more complex nonlinear model, namely a static nonlinearity or a DNN (Figure 1). There are a few reasons for this staged approach. First, it allows us to cast the decoding problem as a classic image enhancement problem that can directly utilize the computer vision literature on super-resolution, in-painting, and denoising. This is especially important for the construction of DNNs, which remain nontrivial to tune for problems in non-standard domains (e.g., image reconstruction from neural spikes). Second, by solving the problem partially with a simple linear model, we greatly reduce the space of transformations that a neural network needs to learn, constraining the problem significantly.

In order to leverage image enhancement tools from deep learning, we need large training data sets. We use an encoder-decoder approach: first, develop a realistic encoding model that can simulate neural responses to arbitrary input images, constrained by real data. We build this encoder to predict the average outputs of many RGCs, but this approach could also be applied to encoders fit on a cell-by-cell basis [3]. Once this encoder is in hand, we train arbitrarily complex decoders by sampling many natural scenes, passing them through the encoder model, and training the decoder so that the output of the full encoder-decoder pipeline matches the observed image as accurately as possible.

## 2.1 Encoder model: simulation of retinal ganglion cell responses

For our encoding model, we create a static simulation of the four most numerous retinal ganglion cell types (ON and OFF parasol cells and ON and OFF midget cells) based on experimental data. We fit linear-nonlinear-Poisson models to RGC responses to natural scene movies, recorded in an isolated macaque retina preparation [7, 10, 12]. These fits produce imperfect but reasonable predictions of RGC responses (Figure 2 A). We averaged the parameters (spatial filter, temporal filter, and sigmoid parameters) of these fits across neurons, to create a single model for each of four cell types. We chose this model as it is simple and a relatively good baseline encoder with which to test our decoding method. (Recently, encoding models that leverage deep neural networks [3, 24] have been shown to fit RGC responses better than the simple model we are using; substituting a more complex encoding model should improve the quality of our final decoder, and we intend to pursue this approach in future work.) To deal with static images, we then reduced these models to static models, consisting of one spatial filter followed by a nonlinearity and Poisson spike generation. The outputs of the static model are equal to summing the spikes produced by the full model over the image frames of a pulse movie: gray frames followed by one image displayed for multiple frames. Spatial filters and the nonlinearity of the final encoding model are shown in Figure 2 B and C.

We then tiled the image space (128 x 128 pixels) with these simulated neurons. For each cell type, we fit a 2D Gaussian to the spatial filter of that cell type and then chose receptive field centers with a width equal to 2 times the standard deviation of the Gaussian fit rounded up to the nearest integer. The centers are shifted on alternate rows to form a lattice (Figure 2 D). The resulting response of each neuron to an example image is displayed in Figure 2 E as a function of its location on the image. The entire simulation consisted of 5398 RGCs.

## 2.2 Model architecture

Our decoding model starts with a classic linear regression decoder (LD) to generate linearly decoded images $I^{LD}$ [37]. The LD learns a reconstruction mapping $\hat{\theta}$ between neural responses $X$ and stimuli images $I^{ST}$ by modeling each pixel as a weighted sum of the neural responses: $\hat{\theta} = (X^T X)^{-1} X^T I^{ST}$. $X$ is augmented with a bias term in the first column. The model inputs are $m$ images, $p$ pixels and $n$ neurons such that: $I^{ST} \in R^{m \times p}, X \in R^{m \times (n+1)}, \hat{\theta} \in R^{(n+1) \times p}$. To decode the set of neural responses $X$ we compute the dot product between $\hat{\theta}$ and $X$: $I^{LD} = X\hat{\theta}$.

The next step of our decoding pipeline enhances $I^{LD}$ through the use of a deep convolutional autoencoder (CAE). Our model consists of a 4-layer encoder and a 4-layer decoder. This model architecture was inspired by similar models used in image denoising [11] and inpainting [35, 22]. In the encoder network $E$, each layer applies a convolution and downsampling operating to the output tensor of the previous layer. The output of the encoder is a tensor of activation maps representing a low-dimensional embedding of $I^{LD}$. The decoder network $D$ inverts the encoding process by applying a sequence of upsampling and convolutional layers to the output tensor of the previous layer. This model outputs the reconstructed image $I^{CAE}$. We optimize the CAE end-to-end through backpropagation by minimizing the pixelwise MSE between the output image of the CAE: $I^{CAE} = D(E(I^{LD}))$ and the original stimuli image $I^{ST}$.

The filter sizes, number of layers, and number of filters were all tuned through an exhaustive grid-search. We searched over the following parameter space in our grid search: number of encoding / decoding layers: [3, 4, 5], number of filters in each layer: [32, 64,128,256], filter sizes: [7x7, 5x5, 3x3], learning rates: [0.00005, 0.0001, 0.0002, 0.0004, 0.0008, 0.001, 0.002, 0.004]. Specific architecture details are provided in Figure 1.

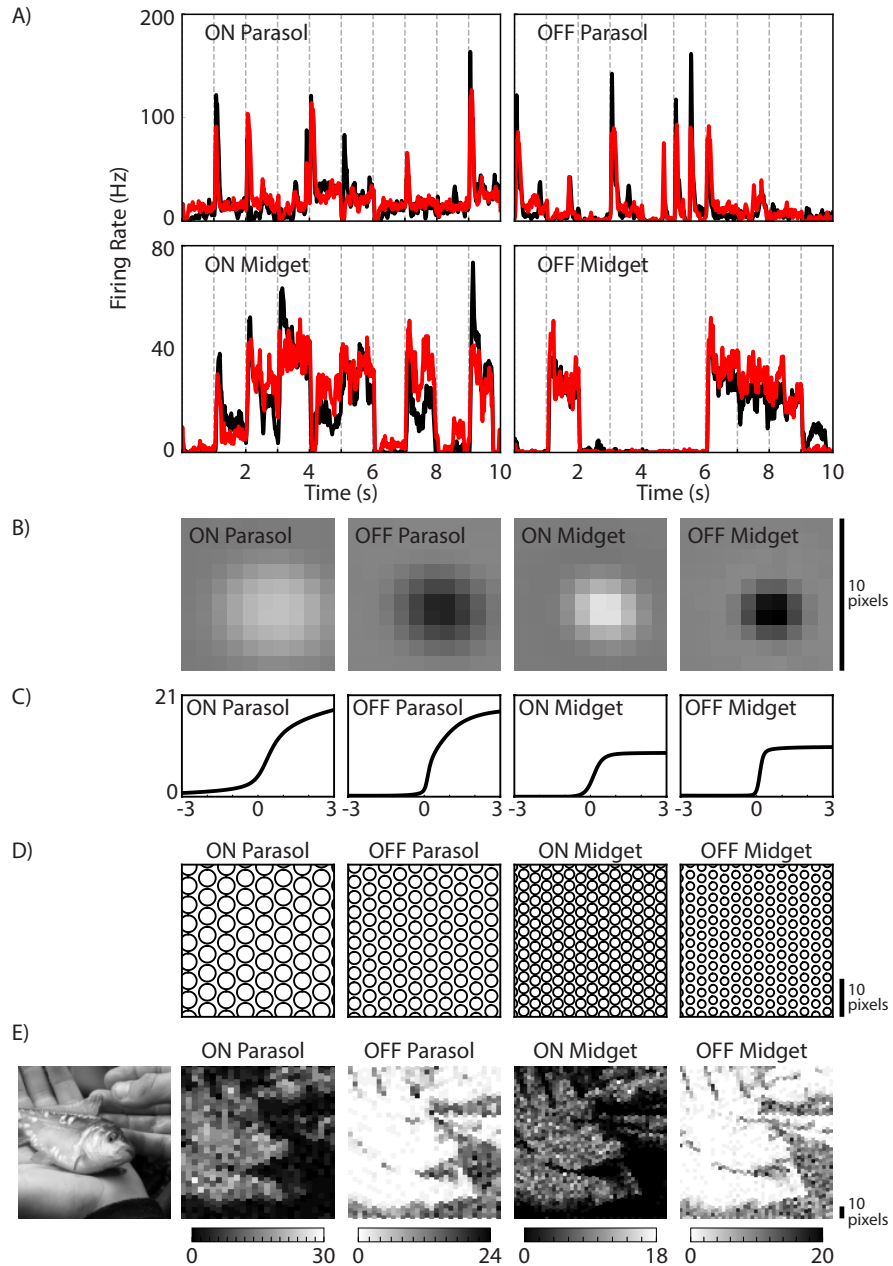

Figure 2: Encoding model. A) Full spatiotemporal encoding model performance on experimental data. Recorded responses (black) vs LNP predictions (red; using the averaged parameters over all cells of each type) for one example cell of each type. The spiking responses to 57 trials of a natural scenes test movie were averaged over trials and then smoothed with a 10 ms SD Gaussian. B) Spatial filters of the simulated neural encoding model are shown for each cell type. C) The nonlinearity following the spatial filter-stimulus multiplication is shown for each cell type. We draw from a Poisson distribution on the output of the nonlinearity to obtain the neural responses. D) Demonstration of the mosaic structure for each cell type on a patch of the image space. The receptive fields of each neuron are represented by the 1 SD contour of the Gaussian fit to the spatial filter of each cell type. E) The response of each cell is plotted in the square around its receptive field center. The visual stimulus is shown on the left. The color maps of ON and OFF cells are reversed to associate high responses with their preferred stimulus polarity.

## 2.3 Training and Evaluation

To train the linear decoder, we iterate through the training data once to collect the sufficient statistics $X^T X$ and $X^T I^{ST}$. We train the convolutional autoencoder to minimize the pixelwise MSE $P_{MSE}$ with the Adam optimizer [15]. To avoid overfitting, we monitor $P_{MSE}$ changes on a validation set three times per epoch and keep track of the current best loss $P_{MSE,best}$. We stop training if we have gone through 2 epochs worth of training data and the validation loss has not decreased by greater than $0.1\% P_{MSE,best}$.

In our experiments we use two image datasets, ImageNet [8] and the CelebA face dataset [21]. We apply preprocessing steps described previously in [17] to each image: **1)** Convert to gray scale, **2)** rescale to 256x256, **3)** crop the middle 128x128 region. From Imagenet we use 930k random images for training, 50K for validation, and a 10k held-out set for testing. We use ImageNet in all but one of our experiments - context-decoding. For the latter, we use the CelebA face dataset [21] with 160k images for training, 30k for validation, and a 10k held-out set for testing.

We evaluate all the models in our results using two separate metrics, pixelwise MSE and multi-scale structural-similarity (SSIM) [36]. Although each metric alone has known shortcomings, in combination, they provide an objective evaluation of image reconstruction that is interpretable and well-understood.

# 3 Results

## 3.1 ImageNet decoding

As expected [33], the linear decoder reconstructed blurry, noisy versions of the original natural images from the neural responses, a result that is attributable to the noisy responses from the RGCs down-sampling the input images. The two-staged model of the CAE trained on the output of the linear decoder (L-CAE) resulted in substantially improved reconstructions, perceptually and quantitatively (Figure 3). L-CAE decoding outperformed linear decoding both on average and for the vast majority of images, by both the $MSE$ and $1 - SSIM$ measures. Qualitatively, the improvements made by the CAE generally show increased sharpening of edges, adjustment of contrast, and smoothing within object boundaries that reduced overall noise. Similar improvement in decoding could not be replicated by utilizing static nonlinearities to transform the linear decoded output to the original images. We used a 6th degree polynomial fitted to approximate the relation between linearly decoded and original image pixel intensities, and then evaluated this nonlinear decoding on held out data. This approach produced a small improvement in reconstruction: 3.25% reduction in MSE compared to 34.50% for the L-CAE. This reveals that the improvement in performance from the CAE involves nonlinear image enhancement beyond simple remapping of pixel intensities. Decoding noisier neural responses especially highlights the benefits of using the autoencoder: there are features identifiable in the L-CAE enhanced images that are not in the linear decoder images (Supplementary Figure 6).

The results shown here utilize a large training dataset size for the decoder so it is natural to ask for a given fixed encoder model, how many training responses do we need to simulate to obtain a good decoder. We tested this by fixing our encoder and then training the CAE stage of the decoder with varying amounts of training data. (Supplementary Figure 8). We observed that even with a small training data set of 20k examples, we can improve significantly on the linear decoder and after around 500k examples, our performances begins to saturate. An analogous question can be asked about the amount of training data required to fit a good encoder and we intend to explore this aspect in future work.

## 3.2 Phase Scrambled Training

A possible explanation for the improved performance of the L-CAE compared to the baseline linear decoder is that it more fully exploits phase structure that is characteristic of natural images [2], perhaps by incorporating priors on phase structure that are not captured by linear decoding. To test this possibility, we trained both linear and L-CAE decoders on phase-scrambled natural images. The CAE input was produced by the linear decoder trained on the same image type as the CAE. Observed responses of RGCs to these stimuli followed approximately the same marginal distribution as responses to the original natural images. We then compared the performance of these linear and

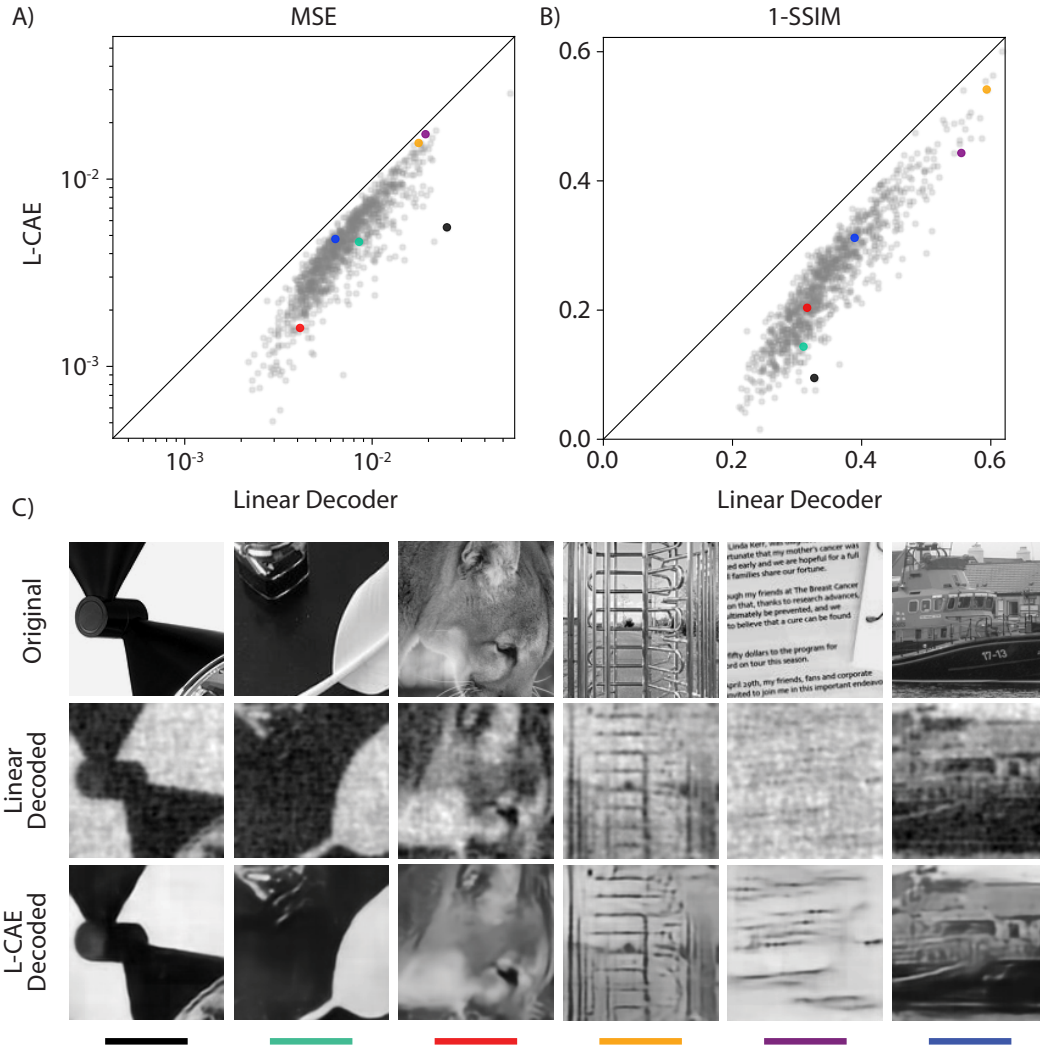

Figure 3: Comparison of linear and CAE decoding. A) MSE on a log-log plot for the ImageNet 10k example test set comparing the L-CAE model trained on ImageNet (only 1k subsampled examples are plotted here for visualization purposes). B) 1-SSIM version of the same figure. C) Example images from the test set show the original, linear decoded, L-CAE enhanced versions. The average (MSE, 1-SSIM) for the linear decoder over the full test set was $(0.0077, 0.35)$ and the corresponding averages for the L-CAE were $(0.0051, 0.25)$.

L-CAE decoders to the performance of the original decoders, on the original natural images (Figure 4). The linear decoder exhibited similar decoding performance when trained on the original and phase-scrambled images, while the L-CAE exhibited substantially higher performance when trained on real images. These findings are consistent with the idea that the CAE is able to capture prior information on image phase structure not captured by linear decoding. However, direct comparisons of the L-CAE and LD trained and tested on phase scrambled images show that the L-CAE does still lead to some improvements which are most likely just due to the increased parameter complexity of the decoding model (Supplementary Figure 7).

## 3.3 Context Dependent Training

The above results suggest that the CAE is capturing important natural image priors. However, it remains unclear whether these priors are sufficient to decode specific classes of natural images as accurately as decoding models that are tuned to incorporate class-specific priors. We explored this in

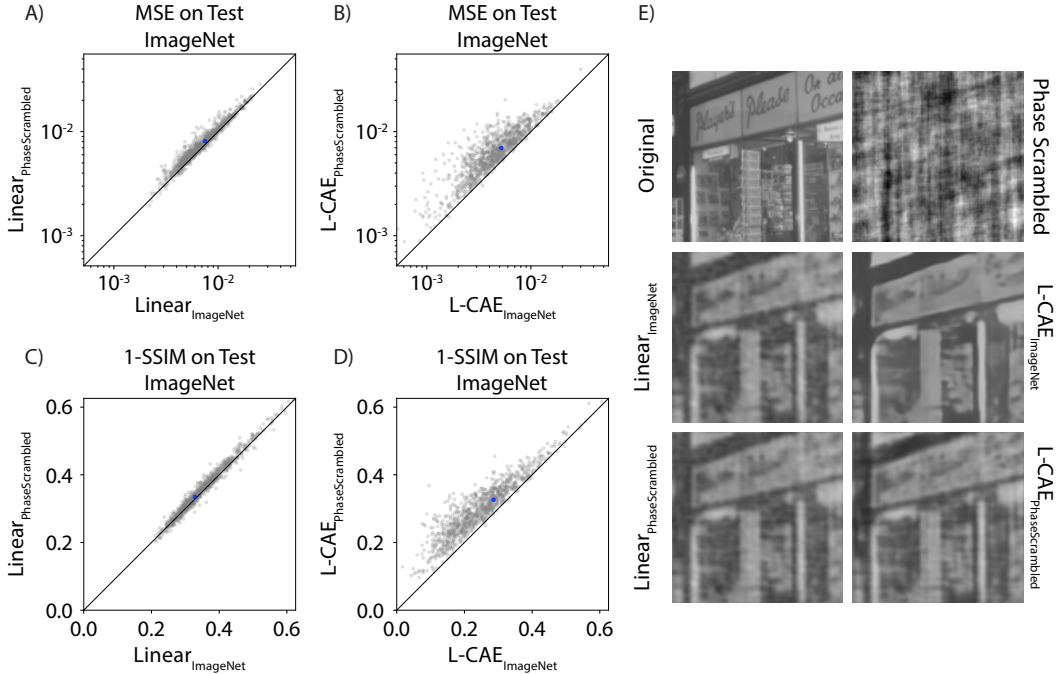

Figure 4: Comparison of phase scrambled and ImageNet trained models. A) MSE on log-log plot comparing the performance of the linear decoder fit on natural images to the linear decoder fit on phase scrambled images. The subscript of each model indicates the dataset on which it was trained. The reported MSE values are based on performance on the natural image test set (1k subsampled examples shown). B) Similar plot to A but comparing the L-CAE fit on natural images to the L-CAE fit on phase scrambled images. C) 1-SSIM version of A. D) 1-SSIM version of B. E) One example test natural image (represented by blue dot in A-D) showing the reconstructions from all 4 models and the phase scrambled version.

the context of human faces by fully re-training a class-specific L-CAE using the CelebA face dataset. Both linear and CAE stages were trained from scratch (random initialization) using only this dataset. As with the phase scrambled comparisons, the CAE input is produced by the linear decoder trained on the same image type. We then compare these different linear decoder and L-CAE models on a test set of CelebA faces. For the linear decoders, we see a 17% improvement in average test MSE and a 14% improvement in 1-SSIM when training on CelebA as compared to training on ImageNet (Figure 5 A and C). We find that the differences in MSE and 1-SSIM between the differently trained L-CAE models are smaller (5% improvement in MSE and a 4% improvement in 1-SSIM) (Figure 5 B and D). The much smaller difference in MSE and 1-SSIM suggests that the L-CAE decoder does a better job at generalizing to unseen context-specific classes than the linear decoder. However, the images show that there are still important face-specific features (such as nose and eye definition) that are much better decoded by the L-CAE trained only on faces (Figure 5E). This suggests that while the natural image statistics captured by the CAE do help improve its generalization to more structured classes, there are still significant benefits in training class-specific models.

## 4 Discussion

The work presented here develops a novel approximate Bayesian decoding technique that uses non-linear DNNs to decode images from simulated responses of retinal neurons. The approach substantially outperforms linear reconstruction techniques that have usually been used to decode neural responses to high-dimensional stimuli.

Perhaps the most successful previous applications of Bayesian neural decoding are in cases where the variable to be decoded is low-dimensional. The work of [5] stimulated much progress in hippocampus and motor cortex using Bayesian state-space approaches applied to low-dimensional (typically

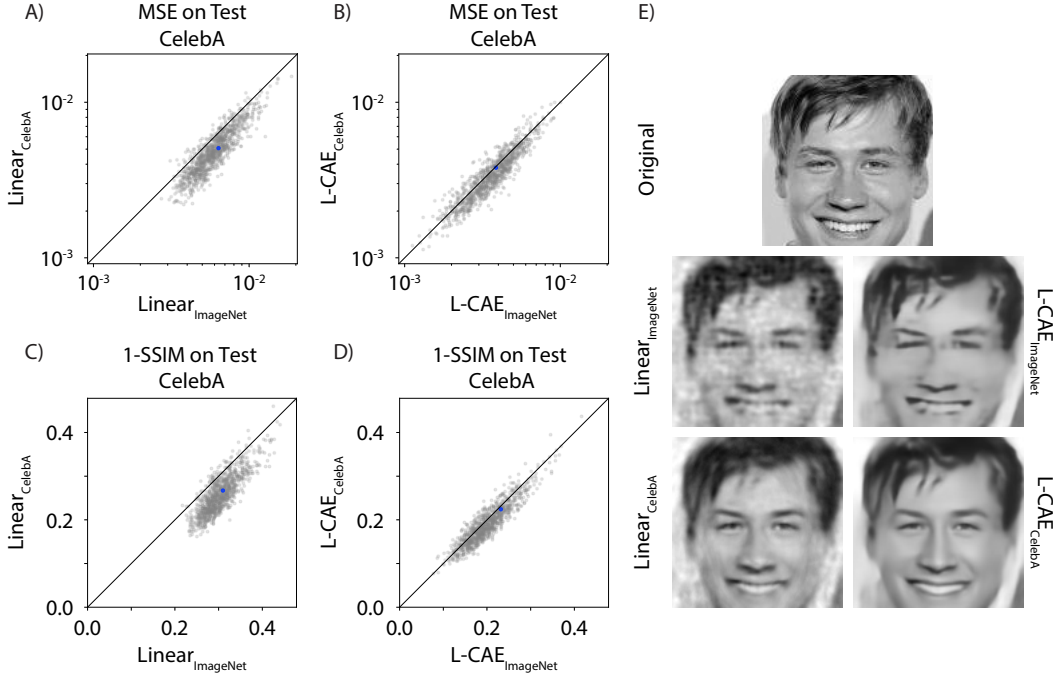

Figure 5: Comparison of CelebA and ImageNet trained models. A) MSE on log-log plot comparing the performance of the linear decoder fit on CelebA to the linear decoder fit on ImageNet. The subscript of each model indicates the dataset on which it was trained. The reported MSE values are based on performance on the natural image test set (1k subsampled examples shown). B) Similar plot to A but comparing the L-CAE fit on CelebA to the L-CAE fit on ImageNet. C) 1-SSIM version of A. D) 1-SSIM version of B. E) One example test natural image (represented by blue dot in A-D) showing the reconstructions from all 4 models.

two- or three-dimensional) position variables; see also [16] and [28] for further details. The low dimensionality of the state variable and simple Markovian priors leads to fast Bayesian computation in these models. At the same time, non-Bayesian approaches based on support vector regression [32] or recurrent neural networks [34] have also proven powerful in these applications.

Decoding information from the retina or early visual pathway requires efficient computations over objects of much larger dimensionality: images and movies. Several threads are worth noting here. First, some previous work has focused on decoding of flicker stimuli [37] or motion statistics [18, 23], both of which reduce to low-dimensional decoding problems. Other work has applied straightforward linear decoding methods [33, 9]. Finally, some work has tackled the challenging problem of decoding still images undergoing random perturbations due to eye movements [6, 1]. These studies developed approximate Bayesian decoders under simplified natural image priors, and it would be interesting in future work to examine potential extensions of our approach to those applications.

While our focus here has been on the decoding of spike counts from populations of neurons recorded with single-cell precision, the ideas developed here could also be applied in the context of decoding fMRI data. Our approach shares some conceptual similarity to previous work [25, 27] which used elegant encoding models combined with brute-force computation over a large discrete sample space to compute posteriors, and to other work [38] which used neural network methods similar to those developed in [41] to decode image features. Our approach, for example, could be extended to replace a brute-force discrete-sample decoder [25, 27] with a decoder that operates over the full high-dimensional continuous space of all images.

Many state-of-the-art models for in-painting and super-resolution image enhancement rely on generative adversarial networks (GANs). However, these models currently require specific architecture tuning based on the exact problem structure. Because our problem involves some complex and unknown combination of denoising, super-resolution, and inpainting, we required a more robust model that could be tested with little hand-tuning. Furthermore, we have no parametric form for the noise

in the linear decoded images, so standard pre-trained networks could not be applied directly. Based on previous work in [39], it seems that autoencoder architectures can robustly achieve reasonable results for these types of tasks; therefore, we chose the CAE architecture as a useful starting point. We have begun to explore GAN architectures, but these early results do not show any significant improvements over our CAE model. We plan to explore these networks further in future work.

In Section 3.3 we saw that even though there were small differences in MSE and 1-SSIM between the outputs of the L-CAE decoders trained on ImageNet vs. CelebA datasets, visually there were still significant differences. The most likely explanation for this discrepancy is that these loss functions are imperfect and do not adequately capture perceptually relevant differences between two images. In recent years, more complex perceptual similarity metrics have gained traction in the deep learning community [42, 20, 13]. While we did not extensively explore this aspect, we have done some preliminary experiments that suggest that using just a standard VGG-based perceptual metric [13] decreases some blurring seen using MSE, but does not significantly improve decoding in a robust way. We plan to further explore these ideas by implementing perceptual loss functions that utilize more of our understanding of operations in the early human visual system [19]. Progress in this space is vital as any retinal prosthetics application of this work would require decoding of visual scenes that is accurate by perceptual metrics rather than MSE.

We have shown improved reconstruction based on simulated data; clearly, an important next step is to apply this approach to decode real experimental data. In addition, we have shown better L-CAE reconstruction only based on one perfect mosaic of the simulated neurons. In reality, these mosaics differ from retina to retina and there are gaps in the mosaic when we record from retinal neurons. Therefore, it will be important to investigate whether the CAE can learn to generalize over different mosaic patterns. We also plan to explore reconstruction of movies and color images.

The present results have two implications for visual neuroscience. First, the results provide a framework for understanding how an altered neural code, such as the patterns of activity elicited in a retinal prosthesis, could influence perception of the visual image. With our approach, this can be assessed in the image domain directly (instead of the domain of spikes) by examining the quality of "optimal" reconstruction from electrical activity induced by the prosthesis. Second, the results provide a way to understand which aspects of natural scenes are effectively encoded in the natural output of the retina, again, as assessed in the image domain. Previous efforts toward these two goals have relied on linear reconstruction. The substantially higher performance of the L-CAE provides a more stringent assessment of prosthesis function, and suggests that the retina may convey visual images to the brain with higher fidelity than was previously appreciated.

## 5 Acknowledgments

NSF GRFP DGE-16-44869 (EB), NSF/NIH Collaborative Research in Computational Neuroscience Grant IIS-1430348/1430239 (EJC & LP), DARPA Contract FA8650-16-1-7657 (EJC), Simons Foundation SF-SCGB-365002 (LP); IARPA MICRONS D16PC00003 (LP); DARPA N66001-17-C-4002 (LP).

## Footnotes

[1]Source Code is available at: https://github.com/nikparth/visual-neural-decode

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
