[Supplementary Material]

# 6 Supplementary Materials

## 6.1 Supplementary Figures

Figure 6: Noisier Neural Responses. We used the same spatial encoding model but divided the firing rate by a factor of 10 before generating Poisson spikes (reducing the input to the Poisson distribution reduces the signal to noise ratio). Randomly chosen decoded images are shown for the original encoder (rows 2+3) and the noisier encoding model (rows 4+5). The L-CAE pulls out features that are not obvious in the noisier linear decoded images.

Figure 7: L-CAE vs linear decoder comparisons for Phase Scrambled and ImageNet. A) MSE comparisons of the L-CAE and linear models trained on ImageNet and tested on ImageNet. B) MSE comparisons of the L-CAE and linear models trained on phase scrambled images and tested on phase scrambled images. The average MSE for $Linear_{PhaseScrambled}$ and $L-CAE_{PhaseScrambled}$ over the full phase scrambled test set was $(0.0102, 0.0075)$ respectively.

Figure 8: Test MSE for L-CAE decoder vs the total amount of training and validation data used. Significant improvements can be achieved up to around 500k training examples - after this, only marginal improvements are gained by increasing training data. We also see that even with a small training + validation dataset of 20k examples, we can still achieve a significant improvement in MSE $(0.0059)$ over the linear decoder MSE $(0.0077)$.