[Reviews · NeurIPS 2017]

Reviewer 1



This paper describes a NN architecture to enhance results of a linear decoder for retinal ganglion cell activity, incorporating natural image statistics to improve the results of the neural decoder. The method is related to recent work in super-resolution, inpainting, and denoising. To produce enough training data, the authors first trained an encoder to model RGC responses (4 kinds of cells, several thousand cells in all). They applied this encoder on images from ImageNet. From these simulated RGC responses, the paper learns a linear decoder of the RGC responses, then trains a CAE to enhance the linearly decoded image. This approach was assessed with pixel-wise MSE and image structural similarity. The paper is clearly written and presents an interesting idea of relevance to both the NN and the visual neuroscience communities. I'm curious about how this architecture would perform at different spatial scales? And if it may be extended to improve decoding performance for other visual features like motion?

Reviewer 2



The paper describes a method to decode natural images from retinal-like activities, using convolutional neural networks. The retinal-like activities are generated by a constructed lattice of linear-nonlinear-Poisson models (separately fitted to RGC responses to natural scenes in a macaque retina preparation) in response to natural static images. After a simple linear decoding of the images from the retinal-like activities, a convolutional neural network further improves on the reconstruction of the original natural images. The paper is clearly written and the results seem sound. A few comments to clarify the motivation, assumptions and impact of the work: -the method proposed is compared to a linear decoder and shown to perform substantially better. However, the performance of the decoding stage will most likely depend on the performance of the encoding stage. A more explicit discussion on the fact that the results of the CAE will ultimately depend on the encoding would clarify the scope and limitations of the study; -on a related note, the authors should be explicit about the reasons for choosing a synthetic encoding stage: why didn’t the authors directly fit their method to the data without the synthetic encoding stage? Too few cells recorded? Too few natural images for fitting the CAE? An attempt at excluding the temporal component of the cells responses, so that the problem reduces to decoding static natural images instead of movies (which would be a more complex task)? This should be more explicitly stated; -in Macintosh et al. 2017 (https://arxiv.org/abs/1702.01825), a convolutional neural network is shown to be a good encoding model of the RGCs in response to natural movies. The authors should discuss their method in the context of such study; -a plot illustrating how the performance of the CAE decoder depends on the amount of data used for training the network would be instrumental for assessing how powerful and practical the approach is; -the authors show that the CAE decoder is able to capture phase structure, unlike the linear decoder. Is the CAE better than the linear decoder mostly because it captures the phase structure? How does the performance between CAE and linear decoder compare in phase scrambled images? It would be helpful to comment on this, and perhaps explicitly plot such comparison. Some typos/minor comments: -lines 83-84, the authors write “resulting response of each neuron…displayed in Figure 3”, however no neural responses are displayed in Figure 3; -line 104, the authors write “all tuned through an exhaustive grid-search”. Providing more details about this tuning would improve the clarity of the paper, and allow for a better assessment of the usability of the method presented; -line 112, the authors write “i = {1,2,…,n}”. However, ’n’ is the number of neurons. Is ’n’ also the number of training epochs? -- After the rebuttal -- The rebuttal has addressed satisfactorily all my concerns/comments, and therefore I would like to increase my score to a clear accept (score: 8).

Reviewer 3



A new LN neural decoding architecture for decoding images using the neural response from a population of retinal neurons is proposed. The N-stage does super-resolution type operation in the natural images context, and implemented by convolutional autoencoder (CAE). The CAE was trained end-to-end to mimic an VAE with a recognition model. The L-stage was simply a least-square decoder. They showed proof of concept on a simulated population (no real data). Two main data sets were used. The example of network performance used ImageNet images, and clearly showed that the two-stage reconstruction method both qualitatively and quantitatively outperformed just linear decoding. A few small examples in sec. 3.1.1-2 remarked that simply adding a nonlinearity to the linear decoder could not math the improvement by using the CAE, and additionally the linear+CAE approach had difficulty reconstructing with phase-scrambled images. These observations led the authors to believe that the CAE component contains more than just a simple nonlinear transformation of pixel intensities, and that is is specifically picking up on phase relationships in the image. The final data set tested the importance of context-dependent training, in which the CelebA face data set was used to see if a CAE trained specifically for face recognition would outperform a generally-trained CAE for image decoding. They found that while the face-trained network did perform better, the improvement was surprisingly small, and that the generally-trained CAE did a good job, suggesting that this type of approach could be a reasonable candidate for general image decoding given enough training data.